# Carrot (*Daucus carota* L.) as Host for *Pentastiridius leporinus* and Phloem-Restricted Pathogens in Germany

**DOI:** 10.3390/biology14091152

**Published:** 2025-09-01

**Authors:** Natasha Witczak, Salma Benaouda, Friederike Wahl, Hendrik Göbbels, Christian Lang, Barbara Jarausch, Michael Maixner

**Affiliations:** 1Agrarservice Hessen-Pfalz GmbH, 67547 Worms, Germany; benaouda@agrarservice-solutions.de (S.B.); goebbels@agrarservice-solutions.de (H.G.); 2Institute for Plant Protection in Fruit Crops and Viticulture, Julius Kühn-Institute, Federal Research Centre for Cultivated Plants, 76833 Siebeldingen, Germany; friederike.wahl@julius-kuehn.de (F.W.); barbara.jarausch@julius-kuehn.de (B.J.); michael.maixner@julius-kuehn.de (M.M.); 3Association of Hessian-Palatinate Sugar Beet Growers e.V., 67547 Worms, Germany; lang@ruebe.info

**Keywords:** Cixiidae, Apiaceae, ‘*Candidatus* Arsenophonus phytopathogenicus’, ‘*Candidatus* Phytoplasma solani’, 16SrXII-P related strain, horticulture, Stolbur, proteobacterium, transmission, host shift, MLST, RFLP

## Abstract

The planthopper *Pentastiridius leporinus* is known as a vector of phloem-restricted pathogens such as ‘*Candidatus* Arsenophonus phytopathogenicus’, ‘*Candidatus* Phytoplasma solani’, and a 16SrXII-P group phytoplasma infecting sugar beet and potato in Germany and causing “Syndrome Basses Richesses” in the former. These pathogens are associated with yellowing, wilting, and low yields. In this study, four pop-up tents were set up in a carrot (*Daucus carota* L.) field located in Bingen on the Rhine, into each of which 20 field-captured female *Pentastiridius leporinus* were released at the beginning of July. After 56 days, nymphs collected from carrot roots were identified by morphological and molecular analyses as *P. leporinus*. The plants inside the tents as well as carrots sampled randomly in the entire field outside the tents were analyzed for infection by the pathogens. For the first time, we observed a direct connection between the typical disease symptoms in carrots, such as leaf yellowing and redness, and the presence of the vector. Our molecular analysis shows that *P. leporinus* can transmit ‘*Ca*. Arsenophonus phytopathogenicus’ and 16SrXII-P related strain to carrots under field conditions. We conclude that *P. leporinus* may play a significant role in the transmission of phloem-restricted pathogens to carrot.

## 1. Introduction

Since 2001, Stolbur-like symptoms have been observed in sugar beet (*Beta vulgaris* ssp. *vulgaris*) associated with the phloem-feeding planthopper *Pentastiridius leporinus* (L., 1761) in France [1,2] and in Germany in the region of Heilbronn (Baden-Württemberg) since 2008 [3]. These symptoms are triggered by three phloem-restricted pathogens that cause the disease “Syndrome Basses Richesses” (SBR) [4]: the γ-proteobacteria ‘*Candidatus* Arsenophonus phytopathogenicus’ (CaAp), the Stolbur phytoplasma ‘*Candidatus* Phytoplasma solani’ (CaPsol; 16SrXII-A), and the 16SrXII-P phytoplasma [5]. Recently, *P. leporinus* has also been observed to transmit these pathogens in potato fields, leading to similar symptoms and severe damage in potato [6,7].

CaAp is the causal agent of SBR in a few European countries [8,9]. It is persistently transmitted by *P. leporinus* and transferred from the females to their offspring through vertical transmission. Both adults and larval stages can transmit CaAp to plants [6]. Although CaAp has recently been detected in onions (*Allium cepa* L.) in Germany, the vector in this case is so far unknown [10].

Phytoplasmas, such as CaPsol, are wall-less prokaryotes of the class Mollicutes [11], and are categorized based on highly conserved 16S ribosomal gene sequences [12]. CaPsol is assigned to the phylogenetic group 16SrXII, subgroup ‘A’. A closely related strain assigned to a new subgroup ‘P’ has been reported from SBR-affected sugar beet as well as from *P. leporinus* and is considered to be a ‘*Ca*. Phytoplasma’ species distinct from CaPsol [5,13]. CaPsol affects numerous plant species, including both wild and cultivated members of Solanaceae, maize, and grapevine, to which it is transmitted from herbaceous hosts by the Cixiid planthopper *Hyalesthes obsoletus*, inducing the Bois noir disease [14]. Different plant species show variable symptoms when infected with CaPsol, which may also be influenced by environmental factors [8]. In contrast to CaAp, phytoplasmas are not vertically transmitted in the vector. Therefore, de novo infection of the vector is required in each generation through feeding on infected plants.

The wide host range of CaPsol includes carrot. Leafhoppers of the genus *Macrosteles* spp. (Hemiptera: Cicadomorpha: Cicadellidae) have been reported as vectors [15,16]. In Spain, CaPsol infection results in high crop losses in carrot [17]. Symptoms, such as small leaves with yellow or red discoloration, deformation, inferior root and shoot growth, and premature senescence, have been described [15,16]. Phytoplasmas of the 16SrΙ group (Aster Yellows) have also been detected in carrots [16,18,19]. For years, carrots have been the highest yielding vegetable crop in Germany. In 2023, 796,700 tons were harvested, and the yield volume increased by 2% compared to the previous year. According to the German Federal Statistical Office, carrots are the third most popular vegetable crop by customers with a cultivated area of 13,500 ha, making up almost 11% of the total area of vegetables grown in the field [20].

Given the economic importance of carrots in Germany and considering the rapid spread of CaAp and group 16SrXII phytoplasmas across many crop species in combination with the host range expansion of *P. leporinus*, there is an urgent need for further research on this complex pathosystem to assess the risk of large-scale epidemics and define effective control strategies to minimize crop losses.

This study was performed to determine if carrot may serve as a host for the planthopper *P. leporinus*, as well as for the phloem-restricted pathogens. In addition, CaPsol genotypes in carrots sampled in the field and from carrots inoculated by *P. leporinus* in transmission trials should be compared. To gain knowledge of host suitability, four experimental tents (1 m^2^) were set up in a carrot field and each inoculated with field-collected female *P. leporinus*. Adults and larvae of *P. leporinus* and carrots (within and outside the tents) were subjected to molecular analysis for the presence of CaAp and CaPsol infections and related pathogen strains using conventional PCR techniques.

## 2. Materials and Methods

### 2.1. Experimental Plot and Design

Sampling of plants and insects as well as transmission experiments using field collected *P. leporinus* were conducted in Bingen on the Rhine (49.951981, 7.928080) in a carrot field adjacent to a potato field infested with *P. leporinus*, CaAp, and CaPsol on the western side. The carrot field was bordered by various landscape structures, including hedges and bushes, on its eastern boundary (Figure 1). The carrots (variety ‘Carvora’) were sown utilizing primed seeds treated with trace elements on 30.04.2024 in double rows on the dams with an inter-row spacing of 60 cm and a sowing depth ranging from 1 and 1.5 cm with an intra-row spacing of 2.5 cm.

### 2.2. Field Collection of P. leporinus and Transmission Experiment

In the observational study, four mesh tents (GardenSkill Ltd., Worcestershire, UK) measuring 1 m × 1 m × 0.65 m were installed at the corners of the experimental plot on 11.06.2024, corresponding to the 37th day after sowing (BBCH 12) (Figure 1), before the emergence of adult *P. leporinus*. The tents were regularly checked for spiders (predators) and closed with cable ties at the zippers to prevent unauthorized access.

*P. leporinus* were collected from the carrot field using a sweeping net shortly after the emergence of adults and were stored in an insect cage. Twenty female individuals were collected for each tent for the transmission trial. Additional specimens were transferred to the lab in an insect cage and frozen at −20 °C on the same day for further analyses.

After an inoculation access period (IAP) of 56 days, the tents were removed for harvesting and to assess the infection of test plants, egg hatching, and larval development. Larvae were collected by harvesting carrots after the absence of adults was confirmed within the tents (Figure 2). All the collected larval instars were transported alive to the laboratory in plastic cups with lids (Nette GmbH, Göttingen, Germany).

The remaining adults and larvae were examined morphologically according to taxonomic keys [21,22] and the larval stage was determined. Larvae were additionally identified by COI [23] barcoding according to the EPPO standard protocol [24]. The sex of adult planthoppers was determined based on the anatomy of their external genitalia.

Carrots from inside the tents and randomly chosen ones from the field were transported to the laboratory for further analyses.

### 2.3. DNA Extraction

DNA was extracted from 150 mg fresh tissue of carrot taproot tips using the Qiagen DNeasy Plant Kit (Qiagen GmbH, Hilden, Germany) according to the manufacturer’s instructions.

In addition, the DNA of the positive controls from the *Catharanthus roseus* master collection of the Julius Kuehn-Institute (JKI) Siebeldingen, Germany, was extracted using a modified CTAB method [25]. For this purpose, 150 mg of the mid-rib of *C. roseus* leaves was ground with 3 mL buffer (Doyle buffer (CTAB (2%, Carl Roth GmbH + Co. KG, Karlsruhe, Germany), TRIS (0.1 M, pH 8, Carl Roth GmbH + Co. KG, Karlsruhe, Germany), EDTA (0.02 M, pH 8, Carl Roth GmbH + Co. KG, Karlsruhe, Germany), PVP-40 (2%, SERVA Electrophoresis GmbH, Heidelberg, Germany), NaCl (1.4 M, Carl Roth GmbH + Co. KG, Karlsruhe, Germany), bidest water (own facility), mercaptoethanol (0.2%, AppliChem, Darmstadt, Germany)). In total, 1.5 mL of extract was incubated at 65 °C for 30 min. The supernatant was transferred to a 2 mL tube after centrifugation for 10 min at 10,000× *g*, and the same volume of chloroform/isoamylalcohol (24:1) (Carl Roth GmbH + Co. KG, Karlsruhe, Germany) was added. The mixture was centrifuged at 10,000× g for 10 min and an equal volume of ice-cold isopropanol (70%, 600 µL) (Carl Roth GmbH + Co. KG, Karlsruhe, Germany) was added to the extract. The preparation was stored at −20 °C overnight and centrifuged the next day at 4 °C for 15 min at 15,000× *g*. The pellet was purified with 70% ethanol (Carl Roth GmbH + Co. KG, Karlsruhe, Germany), dried, and resuspended in 150 µL TE buffer (TRIS (1 M, pH 8, Carl Roth GmbH + Co. KG, Karlsruhe, Germany), EDTA (0.5 M, pH8, Carl Roth GmbH + Co. KG, Karlsruhe, Germany), bidest water (own facility)).

DNA from insects was extracted using the CTAB method [25]. Single specimens were homogenized in 500 µL CTAB buffer (CTAB (2%, Carl Roth GmbH + Co. KG, Karlsruhe, Germany); TRIS (0.1 M, pH 8, Carl Roth GmbH + Co. KG, Karlsruhe, Germany); EDTA (0.02 M, pH8, Carl Roth GmbH + Co. KG, Karlsruhe, Germany); PVP-40 (2%, SERVA Electrophoresis GmbH, Heidelberg, Germany); NaCl (1.4 M, Carl Roth GmbH + Co. KG, Karlsruhe, Germany); bidest water (own facility)) with two metal beads (Werner Ditzinger GmbH, Braunschweig, Germany) per 2 mL microcentrifuge tube (Eppendorf SE, Hamburg, Germany) using a cell mill TissueLyser ΙΙ (Qiagen GmbH, Hilden, Germany) for 1 min at 25 Hz. The samples were incubated for 30 min at 65 °C and 300 rpm (Thermomixer comfort, Eppendorf SE, Hamburg, Germany).

DNA pellets from adult planthoppers were resuspended in 150 µL of TE buffer, while pellets from larvae were resuspended in 30 µL of the same buffer. The DNA content was measured using UV/Vis spectroscopy NP80 (Implen GmbH, Munich, Germany).

### 2.4. Pathogen Detection and Molecular Typing

In total, 166 carrots were analyzed from the carrot field, 97 samples were taken inside the tents, and the remaining 69 carrots were harvested randomly in the field. All plants inside the tents exhibited chlorotic foliage. In contrast, randomized samples collected from outside the tents included both symptomatic and asymptomatic specimens. To detect both pathogens (CaAp and CaPsol), TaqMan assays were performed for quantitative real-time PCR (realtime-PCR) following the protocols in the original papers [7,26].

A total of 62 samples with the highest C_T_ values (10.7 to 32.9) for each pathogen were further analyzed by endpoint PCR with primers Fra4/Fra5 [27], fTuf1/rTuf1, and fTufAy/rTufAy [28] and subjected to molecular typing.

In addition, seven DNA samples of carrots collected from another field at Maxdorf, approximately 80 km southeast of Bingen, were included in the molecular typing. They had been previously tested positive for both CaAp and CaPsol by DLR-RNH (Dienstleistungszentrum Ländlicher Raum Rheinhessen-Nahe-Hunsrück, Bad Kreuznach, Germany) using probe-based realtime-PCR assay for both pathogens [29].

#### 2.4.1. ‘*Ca*. Arsenophonus Phytopathogenicus’ Detection

Endpoint PCR with the primer pair Fra4/Fra5 was carried out to amplify part of the 16S rDNA of CaAp in all samples [27]. For each reaction, template DNA was diluted 1:10 and 2 µL of template were added to a total reaction volume of 20 µL. No template controls (NTC) without template DNA and positive controls were included in each PCR.

CaAp isolate maintained in *Catharanthus roseus* from the master collection of JKI served as positive control (V28.23).

The PCR products were separated on a 1.5% agarose gel (Biozym Biotech Trading GmbH, Vienna, Austria) stained with SERVA DNA Stain Clear G (SERVA Electrophoresis GmbH, Heidelberg, Germany) and visualized using a UV transillumniator QUANTUM ST4 1100 (VILBER, Collégien, France).

#### 2.4.2. ‘*Ca*. Phytoplasma Solani’ Detection

Nested PCR was carried out with the group-specific primer pairs fTuf1/rTuf1 and fTufAy/rTufAy [28] to detect CaPsol in plant and insect samples. PCR conditions corresponded to the original publication, except for annealing temperatures, which were 45 °C for the first and 55 °C for the second round. For nested PCR, 2 µL of the diluted (1:100) PCR product of the first round was used as a template in a total reaction volume of 20 µL.

#### 2.4.3. Molecular Typing

Samples tested positive for CaPsol infection were genetically characterized by restriction fragment length polymorphism (RFLP) and/or multilocus sequence typing (MLST) analyses based on amplified fragments of *tuf*, *stamp*, *secY*, and *vmp1* genes. CaPsol isolates served as positive controls (V44.23 (*tuf*-a), 13299 (*tuf*-b1), 10120 (*tuf*-c)). Amplicons of nested PCR of the *tuf*-gene were subjected to digestion using *Hpa*ΙΙ restriction enzyme (R0171S, New England Biolabs, Ipswich, UK) [30].

The Stolbur-specific *stamp* gene was amplified by nested PCR with the primer pairs Stamp-F/Stamp-R0 and Stamp-F1/Stamp-R1 as described before [31].

The *secY* gene was amplified, using the primer pair PosecF1/PosecR1 followed by PosecF3/PosecR3 as previously described [32]. *Stamp* and *secY* amplicons were further sequenced (Microsynth Seqlab GmbH, Göttingen, Germany).

A fragment of *vmp1* was amplified using the primer pairs StolH10F1/StolH10R1 followed by TYPH10F/TYPH10R [32]. Positive amplicons were subjected to RFLP analysis with *Rsa*Ι restriction enzyme (ER1121, Thermo Fisher Scientific Inc., Darmstadt, Germany) according to the manufacturer’s instructions. The digestion products were separated on a 2% agarose gel stained with SERVA DNA Stain Clear G and a SERVA FastLoad 50 bp DNA ladder (SERVA Electrophoresis GmbH, Heidelberg, Germany) was used to ensure correct evaluation of *vmp1* profiles in the RFLP analysis.

#### 2.4.4. 16S Subgroup Classification of Unclear Samples

Parts of these samples that revealed an unusual restriction profile of the *tuf*-amplicon were subjected to further analysis. The 16S rRNA gene of three samples was amplified by nested PCR with the primer pair P1/P7 [33] followed by R16F2n/R16R2 [34]. Amplicons were sequenced in order to confirm their affiliation to the 16SrXII group and to assign them to a subgroup.

## 3. Results

### 3.1. Symptomatic Distribution in Diseased Carrots

Diseased carrots exhibited Stolbur-like symptoms, including leaf discoloration (yellow and red), deformation, and stunted root growth. Of the 166 carrot samples collected, 97 taproots were obtained from within the four tents and 69 were randomly sampled from outside the tents across the field.

Among the 97 samples within the tents, 21% (20 of 97) exhibited soft consistency, whereas 79% (77 of 97) of the taproots demonstrated firm consistency. All samples displayed yellowing of the foliage. Notably, 2% (2 of 97) of the soft samples exhibited root bipedalisation (Figure 2A).

Of the 69 samples outside the tent, soft consistency of the root was observed in 35% (24 of 69) and firm consistency in 65% (45 of 69). The soft carrot bodies exhibited red coloration (Figure 2C) of the foliage in 25% (6 of 24) of cases, with some instances progressing to necrosis, of which 50% (12 of 24) were analyzed, and two-thirds of these were infected with CaPsol (16SrXII-A). Yellowing of the foliage (Figure 2B) was observed in 50% (12 of 24) of the samples, and one specimen exhibited bipedalization. In addition, 25% (6 of 24) of plants showed no leaf discoloration. Among the firm carrots outside the tents, 2% (1 of 45) exhibited red discoloration and loss of foliage, 78% (35 of 45) displayed yellow discoloration, and 18% (8 of 49) showed no foliar symptoms.

### 3.2. Association of P. leporinus with Carrot and Pathogens

The year 2024 was characterized by extreme temperature fluctuations and heavy rainfall events (https://www.wetter.rlp.de/Agrarmeteorologie/Wetterdaten/Rheinhessen, accessed on 3 December 2025) and led to a much later flight of *P. leporinus* paired with a short and intense flight period with regional variation. During the flight period, 53 adult *P. leporinus* (30 ♀♀, 23 ♂♂) were caught from carrot outside the tents, of which 40 (23 ♀♀, 17 ♂♂) were analyzed.

After an inoculation access period (IAP) of 56 days, the tents were removed. A total of 19 larvae in various larval stages were collected during harvest from carrots under the tents (Figure 3), of which ten larvae had already reached the 5th larval stage.

Adults and larvae were identified as *P. leporinus* according to morphological determination. The identity of five larvae with *P. leporinus* was confirmed by molecular barcoding. COI sequences were between 99.84 and 100% homologous to sequences of *P. leporinus* deposited in the BOLD database (MZ632211; Barcode of Life Data Systems v3, https://v3.boldsystems.org/, accessed on 17 February 2025).

#### 3.2.1. Pathogens and Molecular Typing in Plant Material

A total of 166 carrot samples from Bingen were analyzed using realtime-PCR, comprising 97 samples from within tents and 69 from open field. Of these 166 samples, 62 samples exhibiting C_T_ values from 10 to 32.5 for both pathogens were subsequently examined and characterized by endpoint PCR. Furthermore, seven samples from DLR-RNH from Maxdorf were subjected to endpoint PCR analysis.

A proportion of 31% (30 of 97) of carrot samples from the tent were double infected with both pathogens, while 41% (40 of 97) of the samples were infected by CaAp only, and 3% (3 of 97) by CaPsol only. This result indicates that *P. leporinus* successfully transmitted both pathogens to carrots under field conditions. The double infection rate was considerably higher in the open field (83%, 57 of 69). Single CaAp and CaPsol infections were detected in 9% (6 of 69) and 7% (5 of 69) of the samples, respectively. Only 1% (1 of 69) of the samples tested negative according to the realtime-PCR assays (Table 1).

Among the 62 subsamples with the lowest C_T_ values, 36 of 97 (34.9%) were obtained from the tents, while 26 of 69 (17.9%) were collected from the field. Within the tents, a single CaAp infection was detected in 50% (18 of 36) of the subsamples. No single CaPsol (16SrXΙΙ-A) or double infection was analyzed (Table 2).

A proportion of 50% (13 of 26) of the carrot subsample from the field tested positive for single CaAp infection. Four samples (15%) showed a double infection which could be assigned to 16SrXΙΙ-A, *tuf*-type ‘b1′ (KJ469708) (Table 2), while none of the samples obtained a single CaPsol infection. Negative tested material, amounting to 8% (2 of 26), was also detected (Table 2).

Almost 57% (4 of 7) of positive samples provided by DLR-RNH were infected by CaPsol, while one sample was infected by both pathogens. The samples could be assigned to 16SrXΙΙ-A, *tuf*-type ‘b1′ (Table 2).

All CaPsol-positive samples subjected to MLST exhibited an identical *stamp* genotype that showed a 100% match with *stamp* genotype ‘FR662/23’ obtained from sugar beet in northern France (PP731988) [9]. Sequencing of the *secY* amplicons revealed a 100% match with *secY*-C (JQ977709) in all seven samples, which had previously been detected in field in *Convolvulus arvensis* [35], *Vitis vinifera* [36], and *Reptalus artemisiae* (syn. *R. quinquecostatus*) [37], as well as *H. obsoletus*, *P. leporinus*, *Beta vulgaris* ssp. *vulgaris*, and *Solanum tuberosum* ssp.* tuberosum* (Witczak et al., unpublished). Regarding the *vmp1* genotype, only V4 (KJ469730) was detected, with one exception of V14 (KJ469732) in the samples from Maxdorf (Table 2).

#### 3.2.2. Detection of Pathogens in *P. leporinus*

Forty adult *P. leporinus* individuals from open field and 19 nymphs from the tents were tested for the presence of CaAp and CaPsol. Of the 40 adults tested, 55% (22; 12 ♀♀, 10 ♂♂) gave positive results for CaAp compared to 89% (17 of 19) of the larvae. From 22 CaAp-infected adult *P. leporinus* and eight larvae, which were tested negative for CaPsol-infection by endpoint PCR, *tuf*-PCR resulted in the amplification of a product whose digestion profile differed from the known CaPsol *tuf*-type patterns. Sequencing of the *tuf*-amplification product showed 99.12% sequence identity with CaAp isolate Z5 (PP950434) from onion [10]. Thus, the *tuf*-gene of CaAp was amplified in nested PCR, too.

No exclusive CaPsol infection was detected in planthoppers (adults and larvae), precluding characterization by RFLP and MLST. No infection by any of the three pathogens was detected in 40% (16 of 40 adults) and 11% (2 of 19 larvae) of the cases (Table 3).

#### 3.2.3. Detection of 16SrXΙΙ-P Related Phytoplasma in *P. leporinus* and *D. carota*

In addition to CaAp and CaPsol, all samples were also analyzed for 16SrXΙΙ-P using nested PCR with primer pairs fTuf1/rTuf1 and fTufAy/rTufAy followed by digestion with *Hpa*ΙΙ restriction enzyme.

Among the 40 adult *P. leporinus*, 16SrXΙΙ-P double infection with CaAp was detected in two female specimens (5%). No 16SrXΙΙ-P was found in the 19 larvae examined.

In the carrot samples, a single 16SrXΙΙ-P related infection was detected in 3% (1 of 36) and a co-infection with CaAp in 19% (7 of 36) inside the tents of the carrot subsamples. Outside the tents, 16SrXΙΙ-P related strain was also identified in the subsamples. A single infection was observed in 4% (1 of 26) and a co-infection in 23% (6 of 26) of the samples (Table 3).

In the nested PCR for TufAy, a fragment of 940 bp was amplified for the common CaPsol *tuf*-genotypes (*tuf*-a, -b1, and -c) and for the 16SrXII-P related strain. A typical band pattern was found for the 16SrXΙΙ-P type by digestion with *Hpa*ΙΙ (Figure 4). No fragment was amplified for 16SrXΙΙ-P phytoplasma by MLST using the specific primers [31,32]. To confirm the correspondence of the specific *tuf*-restriction profile with 16SrXII-P phytoplasma, the 16S rRNA gene of these samples was amplified by nested PCR with the universal primer pairs P1/P7 [33], followed by R16F2n/R16R2 [34], and the amplicons were sequenced (Acc. No.: PX113210, PX113211, PX113212, PX113213). Comparisons show that they are most similar to the 16SrXII-P phytoplasma.

The positive 16SrXΙΙ-P sequences were compared in NCBI (National Library of Medicine, https://www.ncbi.nlm.nih.gov/, accessed on 16 December 2025) with all sequences in the database. Based on the 16S rRNA gene, one of the sequences (Acc. No.: PX113213) matched 100% with the CaPsol isolate 916/22 (OQ717667). The remaining sequences exhibited a similarity ranging from 99.80% to 99.91% when compared to the latter.

#### 3.2.4. Sensitivity of Realtime-PCR and Nested PCR Protocols

A subsample of 62 carrots with the highest C_T_ values for CaAp and CaPsol were selected to further characterize positively tested samples (Table 1 and Table 3). The samples were subjected to conventional (nested) PCR testing for both pathogens. The realtime-PCR results diverged substantially from the results of nested PCR. Notably, the detection of CaPsol was particularly challenging via nested PCR (Figure 5).

A confusion matrix was employed to compare the realtime-PCR and endpoint PCR methodologies for each pathogen in order to compare results and highlight similarities or differences.

Regarding the detection of CaAp infection (Figure 5A), realtime-PCR results were highly reliable. In total, 48 out of 62 samples were positive by both methods; in this case, both methods corresponded. Four samples were positive by realtime-PCR, but negative by conventional PCR, indicating differences in methodology. No samples were negative by realtime-PCR and positive by endpoint PCR. Ten samples were negative by both methods; here, the results of both methods matched.

Regarding the detection of CaPsol (Figure 5B), however, nested PCR results were much less reliable. Only 4 out of 42 positively tested samples matched with the realtime-PCR results. In total, 38 of 62 samples were positive by realtime-PCR but not confirmed by nested PCR. Twenty samples tested negative in both methods.

Regarding 16SrXII-P related infection (Figure 5C), the same realtime-PCR results were used as for CaPsol. In total, 15 out of 42 positively tested samples were detected by both methods in this context. Twenty-seven samples were found to be infected with CaPsol by realtime-PCR, which is not consistent with the nested PCR results. Twenty samples tested negative in both methods, which is consistent with the result from Figure 5B.

## 4. Discussion

*Pentastiridius leporinus* is the main vector of ‘*Ca*. Arsenophonus phytopathogenicus’ (CaAp) and phytoplasmas of group 16SrXII, which infect two important arable crops, sugar beet and potato [1,2,3,5,6,7,13]. Infected crops serve as a source of infection for immature vectors, which can migrate as already infectious adults into new, healthy fields in the subsequent year [6]. *P. leporinus* initially switched hosts from reed (*Phragmites australis*) to sugar beet (*Beta vulgaris* ssp. *vulgaris*) [38], with a recent expansion to potato (*Solanum tuberosum* ssp. *tuberosum*) [6]. Due to its adaptability to crop rotations, such as sugar beet–winter wheat [39] and now potato–winter wheat or sugar beet–potato, along with its rapid spread and occasionally high population densities, *P. leporinus* is proving to be an effective vector for three pathogens to those crops. This has facilitated the extension of the host range of the pathogens, particularly CaAp.

Carrots represent a crucial vegetable crop in Germany, consistently yielding the highest production (796,700 t in 2023) with increasing yields. Its cultivation area, approximately 13,500 ha, is significant for vegetable production, comprising approximately 11% of the total vegetable growing area. Observations of infected carrot plants resembling Stolbur infection prompted inquiries into whether carrots serve as hosts for *P. leporinus*. Furthermore, it is necessary to ascertain whether *P. leporinus* transmits CaAp and/or CaPsol or 16SrXII phytoplasma to carrots, and to identify the specific genotypes of CaPsol transmitted by this or other vectors.

The observations of the feeding and oviposition of the adult females on carrot as well as the development of the larvae up to the final larval stage (5th stage) on the carrot roots suggest that the carrot probably serves as a food and host plant for *P. leporinus* and that the planthopper was able to expand its host range.

Potential factors contributing to the host range expansion of *P. leporinus* may include phytochemical compatibility, wherein a chemical similarity exists between the historical host and the novel hosts. The favorable chemical composition of the new hosts would then have enabled the successful invasion of *P. leporinus* into arable and vegetable crops [40]. Conversely, adaptations of the vector might have facilitated the expansion of its host range, for instance, by exhibiting host-specific gene expression profiles [41]. Furthermore, environmental changes and anthropogenic interventions, such as the loss of natural habitats [42], could have contributed to a selective pressure towards host range expansion.

To evaluate the transmission potential of *P. leporinus* while excluding other vectors, experimental tents were set up in a carrot field and female *P. leporinus* were placed into the cages. In a study involving 166 harvested carrots, all three pathogens were identified. The prevalence of a single CaAp infection was 27.7% (49 of 166), with a higher incidence observed within the tents than in the field (Table 1). The prevalence of a single CaPsol infection was 4.8% (8 of 166) across the entire sample, with a slightly higher prevalence in the field than in the tents. The highest prevalence, 52.4% (87 of 166), was observed in the double-infected carrot samples, with a higher prevalence of double infection detected in the field samples. The increased infection rates outside the tents suggested an uncontrolled influx of various potential vectors and/or immigration of infected *P. leporinus*, in contrast to the controlled presence of a single vector (*P. leporinus*) within the tents. The elevated infection rates for CaAp indicate that *P. leporinus* is an effective vector for CaAp, as evidenced by the prevalence of individuals, which is consistent with the findings from potato studies [6]. CaAp can be transmitted both vertically and horizontally [38], whereas CaPsol can only be transmitted horizontally.

Unspecific banding patterns in the digestion of certain *tuf*-amplicons prompted further investigation, revealing the presence of 16SrXII-P related strain. This discovery led to the identification of a double infection with 16SrXII-P related phytoplasma in two adults (Table 3). Using RFLP analysis of the *tuf* gene, it is feasible to distinguish between 16SrXII-A and 16SrXII-P as well as similar strains. However, the efficacy of this method in differentiating the 16SrXII subgroup remains uncertain, which, if effective, would facilitate comparisons within the Stolbur group. A banding pattern based on the digestion of the *tuf*-gene documented for ‘*Ca*. Phytoplasma convolvuli’ (16SrXII-H) [43] is distinct from 16SrXII-A and 16SrXII-P related strain (Appendix A). Further comparative analyses of subgroups of the Stolbur group are necessary to evaluate whether this differentiation method is a viable approach for identifying subgroups within the Stolbur group. To date, the pathogen 16SrXII-P has been identified in both sugar beet and *P. leporinus* [5,13]. In addition to CaAp, the 16SrXΙΙ-P related strain was detected in planthopper and carrot samples collected from within the tents. Since the tents excluded other vectors, these observations suggest a specific association between the pathogen 16SrXII-P related phytoplasma and *P. leporinus*. This hypothesis is corroborated by the initial description of 16SrXII-P in sugar beet [5], with *P. leporinus* being the only identified vector to date [13] as well as reported in this study.

Comparative analyses of the subsample results of realtime-PCR and nested PCR indicated that both methods do not differentiate between 16SrXII-A and 16SrXII-P nor similar strains (Table 1, Figure 5). This limitation results in analytical distortions, and may lead to erroneous assumptions and interpretations. Recent findings underscore the necessity to consider these pathogens (16SrXII-A and 16SrXII-P) separately [13]. It is crucial to adapt the primers for realtime-PCR to each specific pathogen to ensure accurate collection of data regarding infection frequencies, among other metrics. In the absence of such adaptations, conventional methods followed by RFLP analysis or sequencing must be employed to ensure accurate conclusions. It is plausible that previous studies may have misclassified samples as positive for CaPsol when they were positive for 16SrXII-P phytoplasma.

16SrXII-A (*tuf*-b1) was detected along with CaAp and the 16SrXII-P related strain in carrots. The transmission of 16SrXII-A may involve other vectors such as *H. obsoletus*, which are present in Bingen. MLST analysis indicated similarities among the genotypes. The *stamp* genotype ‘FR662/23’ (PP731988) has thus far only been identified in sugar beet in France, suggesting its origin [9]. In addition, carrot samples from the DLR-RNH tested positive for CaPsol. No further instances of *stamp* genotype ‘FR662/23’ have been reported in the literature to date. Both the *vmp1* genotypes V4 and V14 were classified within the bindweed-related cluster. Based on the study of Quaglino et al. (2016), V4 has been detected exclusively in hosts such as *Vitis vinifera* [44] and bindweed (*Convolvulus arvensis*) [35], as well as in various vectors, including *H. obsoletus*, *Anaceratagallia ribauti* [43], and *Reptalus* spp. [37]. V14 has been identified in *V. vinifera* [45], *C. arvensis*, and *R. panzeri* [37].

In summary, the rapid expansion of the host range of *P. leporinus* raises several questions. It remains unclear whether the species has evolved from an originally monophagous or oligophagous feeding habit on reeds to a polyphagous diet on cultivated plants, or whether it has always been polyphagous and had been overlooked because of its small population size and lack of vector properties [46]. These open questions emphasize the need for further research on the ecology of *P. leporinus* to better understand its role as a vector and to develop appropriate management strategies.

Extending the host range of pathogens (CaAp and 16SrXII phytoplasmas) to carrots is a significant development that could have far-reaching consequences for carrot cultivation in Germany. Given the economic importance of carrots in German vegetable production [20], further studies are needed to fully understand the economic and epidemiological consequences of these infections. The observed physiological changes, particularly the soft consistency of some carrots, indicate potential problems for marketing and processing, similar to sugar beet [47] and potato [6]. Therefore, it is crucial to develop prevention and management strategies to ensure the quality and productivity of carrot production in Germany and minimize the potential negative impacts on the industry.

## 5. Conclusions

This study provides evidence that *P. leporinus*, known as a vector of phloem-restricted pathogens, specifically ‘*Ca*. Arsenophonus phytopathogenicus’ (CaAp) and 16SrXII phytoplasmas, in arable crops such as sugar beet and potato, is also associated with carrot. Results of enclosure experiments with carrots and adult *P. leporinus* prove that *P. leporinus* deposits eggs and is able to complete a substantial part of its larval development on carrot. It is further shown that the vector was able to transmit CaAp as well as a group 16SrXII-P phytoplasma to carrot. These results indicate a further host range expansion of *P. leporinus* and the associated pathogens. It will be necessary to further investigate a possible epidemiological role of cultivated or wild carrot for the *P. leporinus*-associated pathosystem. The results of this study confirm the close association between *P. leporinus* and CaAp and underscore the importance of differentiating between CaPsol and 16SrXII-P phytoplasmas in epidemiological studies. Further studies are necessary to identify the drivers of the current *P. leporinus* host range expansion.

## Figures and Tables

**Figure 1 biology-14-01152-f001:**
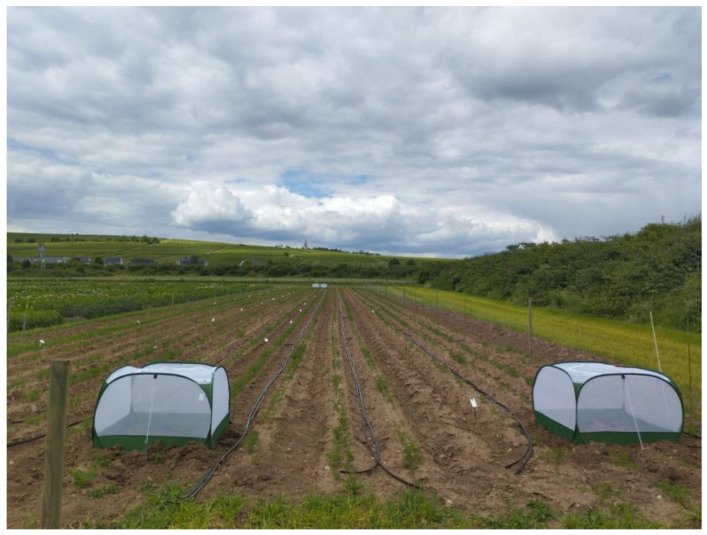
Carrot field during tent installation. To the left is the adjacent potato field, and to the right are various landscape structures. The tents were established above two plant rows (Photo, H.G.).

**Figure 2 biology-14-01152-f002:**
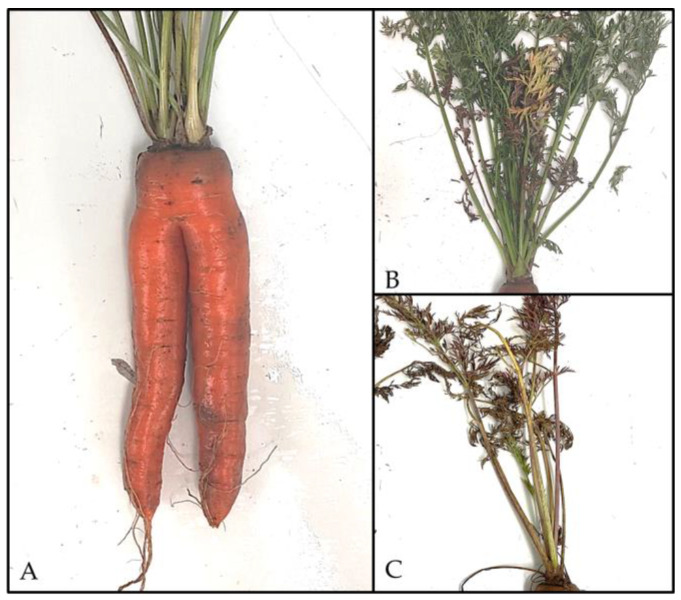
Symptomatic carrots from the field trial. (**A**) Carrot taproot with bipedalism. The foliage of some samples showed (**B**) yellowing or (**C**) red discoloration. (Photos: S.B.).

**Figure 3 biology-14-01152-f003:**
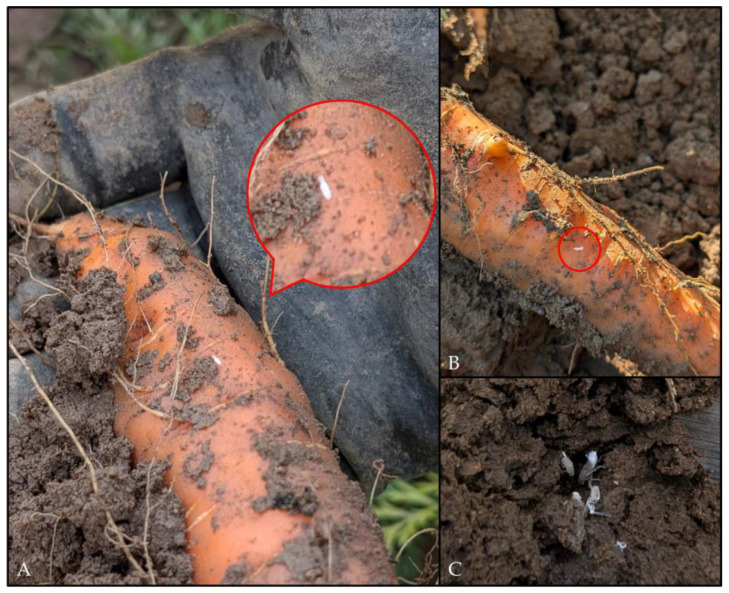
*P. leporinus* larvae found on carrot plants in Bingen on the Rhine in September 2024. (**A**,**B**) Larvae on carrot roots; red circles show larvae. The carrots exhibited deformities and beardedness. (**C**) In addition to young larval stages (**A**,**B**), older larvae were also found in carrot. (Photos: N.W.).

**Figure 4 biology-14-01152-f004:**
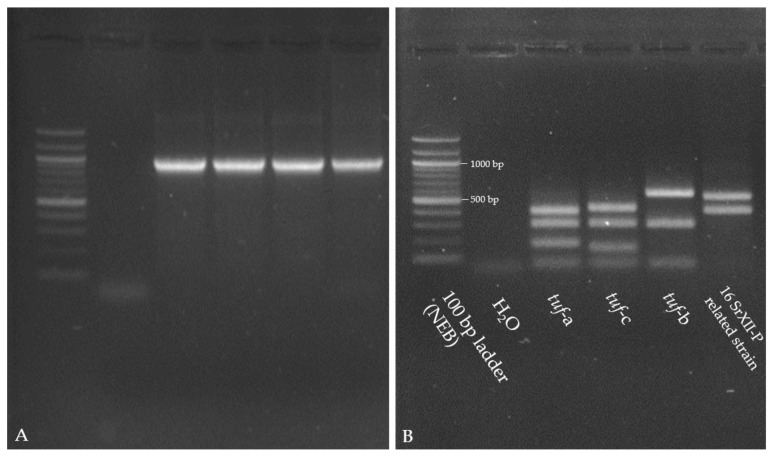
Gel images 1.5% (**A**) and 2% (**B**) agarose gels; both stained with SERVA DNA Stain Clear G of *tuf*-amplicons for CaPsol reference strains (*tuf*-a: V44.23; *tuf*-c: 10120; *tuf*-b1: 13299), 16SrXΙΙ-P related strain, and 100 bp ladder (NEB). (**A**) Gel image of amplicons from nested PCR with the primer pairs fTuf1/rTuf1 and fTufAy/rTufAy [28]. All amplicons had an expected fragment size of 940 bp. (**B**) Restriction fragment profiles of the amplicons digested with *Hpa*ΙΙ. For 16SrXΙΙ-P, the specific banding pattern is 550 bp and 400 bp.

**Figure 5 biology-14-01152-f005:**
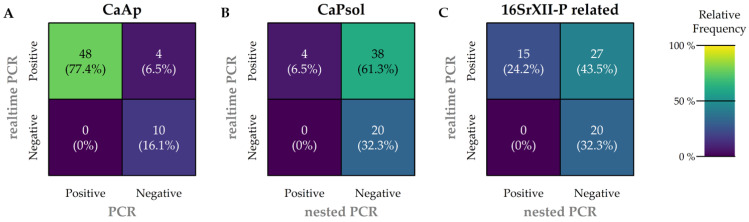
Confusion matrices were constructed for all three pathogens to compare the realtime-PCR method with conventional nested PCR results from the subsamples. The frequencies on the main diagonal (**top left** to **bottom right**) indicate the degree of concordance between the realtime-PCR and conventional nested PCR results, with higher frequencies signifying greater agreement between the methods. A match is present if both methods provide positive results (**top left**) and/or if both methods provide negative results (**bottom right**). A mismatch occurs if the nested PCR is negative but the realtime-PCR is positive (**top right**) and/or if the nested PCR is positive but the realtime-PCR is negative (**bottom left**). (**A**) CaAp realtime-PCR compared to endpoint PCR detection of CaAp. (**B**) CaPsol realtime-PCR test results compared to CaPsol nested PCR results. (**C**) Using the same CaPsol realtime-PCR test results as in (**B**), but compared to 16SrXII-P related nested PCR results with subsequent digestion.

**Table 1 biology-14-01152-t001:** Prevalence of CaAp, CaPsol (16SrXΙΙ-A), and double infection in total carrot samples categorized into tents and open field were analyzed by realtime-PCR assays for CaAp and CaPsol.

Sample	Single CaAp Prevalence	Single CaPsol Prevalence	Double Infection CaAp-CaPsol Prevalence
Carrots (tents)	41% (40/97)	3% (3/97)	31% (30/97)
Carrots (field)	9% (6/69)	7% (5/69)	83% (57/69)

**Table 2 biology-14-01152-t002:** Prevalence of CaAp, CaPsol (16SrXΙΙ-A), and double infection in carrot subsample (N = 62) of the 166 samples shown in Table 1 with C_T_ values of 10 to 32.5 and samples from DLR-RNH from Maxdorf analyzed with conventional PCR and CaPsol multilocus genotypes.

Scheme 1.	Single CaAp Prevalence	Single CaPsol Prevalence	Double Infection CaAp-CaPsol Prevalence	CaPsol *tuf*/*stamp*/*secY*/*vmp1* Genotype	Acc. No. *tuf*/*stamp*/secY/*vmp1*
Carrots (tents)	50% (18/36)	0%	0%	-	-
Carrots (field)	50% (13/26)	0%	15% (4/26)	*tuf*-b1/FR662_23/C/V4	KJ469708/PP731988/JQ977709/KJ469730
Carrots (DLR-RNH)	0%	57% (4/7)	14% (1/7)	*tuf*-b1/FR662_23/C/V4 *tuf*-b1/FR662_23/C/V14	KJ469708/PP731988/JQ977709/KJ469730 KJ469708/PP731988/JQ977709/KJ469732

**Table 3 biology-14-01152-t003:** Prevalence of 16SrXΙΙ-P related strain and double infection in adults, larvae, and the subsample of total analyzed carrots analyzed in this work with RFLP (Figure 4) and sequencing with the universal primer pairs P1/P7, followed by R16F2n/R16R2.

Sample	Single 16SrXΙΙ-P Related Strain Prevalence	Double Infection CaAp-16SrXΙΙ-P Related Strain Prevalence
Adult *P. leporinus* (field)	0%	5% (2/40)
*P. leporinus* larvae (tents)	0%	0%
Carrots (tents)	3% (1/36)	19% (7/36)
Carrots (field)	4% (1/26)	23% (6/26)
Carrots (DLR-RNH)	29% (2/7)	0%

## Data Availability

The original contributions presented in the study are included in the article, further inquiries can be directed to the corresponding author.

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
