# Peer review of "Carrot (Daucus carota L.) as Host for Pentastiridius leporinus and Phloem-Restricted Pathogens in Germany"

_biology, 2025, doi:10.3390/biology14091152_

Round 1

Reviewer 1 Report

Comments and Suggestions for Authors
  • Line 155 specifies the number of samples collected, but their nature remains unclear. Please clarify: Were all samples exclusively from symptomatic plants, or did the collection include asymptomatic (healthy-appearing) specimens as well? In line 228, it is understood that all samples are symptomatic, but it should be stated in the method section that the sampling was made from symptomatic plants.
  • Were the PCR conditions identical for both the fTuf1/rTuf1 and fTufAy/rTufAy primer pairs? I recommend adding the specific cycling parameters for the tuf primers under the subheading in line 179 ('PCR conditions for tuf primers') to ensure methodological transparency. This addition would facilitate experimental reproducibility, particularly given that these primers target the same gene but may require different amplification conditions.
  • The statement in line 194 regarding additional PCR testing of STAMP-positive samples with SecY/VMP1 primers appears redundant. If the stamp gene had been negative, wouldn't verification have been done through vmp1 and secy genes? this sentence should be removed to streamline the text.
  • The heading in line 220 can be corrected as follows; symptomatic distribution in diseased carrots
  • Please describe the sex-determination methodology (morphological/molecular) for captured insects in the Methods section. The reported sex ratios (line 245) lack this critical experimental detail.
  • The parenthetical expressions at the end of the figure caption on lines 226 and 252 should be removed or corrected to "photos" instead of "fotos".
  • While line 245 reports the sex ratio of captured insects, the Methods section lacks details on how males and females were distinguished.
  • Line 267 reports amplification of 'Ca. A. phytopathogenicus' (CaAp) tuf gene via nested PCR, but the Methods section (subsections 2.4.1 and 2.4.3) fails to specify: The primer pairs used for CaAp tuf amplification, Whether these were identical to those used for 'Ca. P. solani' (CaPsol), and If different, the rationale for primer selection. If the same primer pairs successfully amplified both 'Ca. P. solani' (Stol) and 'Ca. A. phytopathogenicus' (CaAp) tuf genes in nested PCR, this finding is particularly noteworthy, as these primers were previously considered Stol-specific.
  • Since the two sentences below Table 1 are explanatory for Table 1, the space between lines 295 and 296 should be removed.
  • The accession number should be written in parentheses on line 347.
  • A more symmetrical image can be obtained if the dimensions of the two different agarose gel images in line 348 are equalized.
  • On line 357, the genbank accession number should be written in parentheses.
  • In Figure 6 in Appendix a, the first well is left blank. A marker should be written on the next well, and the characteristics of this marker should be specified in the figure caption. Furthermore, the in silico gel figure is unnecessarily long; the figure length should be shortened by cutting off the top.

Author Response

For research article

Response to Reviewer#1 Comments

Sunday, 10.08.2025

Journal: Biology (ISSN 2079-7737)

Manuscript ID: biology-3820481

Type: Article

Title: Carrot (Daucus carota L.) as Host for Pentastiridius leporinus and Phloem Restricted Pathogens in Germany

Authors: Natasha Witczak *, Salma Benaouda, Friederike Wahl, Hendrik Göbbels, Christian Lang, Barbara Jarausch, Michael Maixner

Section: Microbiology

Special Issue: Phytoplasmas: Molecular Characterization and Host–Pathogen Interactions (2nd Edition)

We thank the reviewers for taking the time to read our paper and for their helpful suggestions and comments. These have greatly improved our work. We have carefully addressed all their feedback during the revision. We have explained the changes we made for each comment. Our detailed responses to each point are below.

Answer to Reviewer #1

  • Comment: Line 155 specifies the number of samples collected, but their nature remains unclear. Please clarify: Were all samples exclusively from symptomatic plants, or did the collection include asymptomatic (healthy-appearing) specimens as well?
  • Answer: All plants exhibited chlorotic foliage within the tents. Among these, 77 of the 97 plants demonstrated firm consistency (edited document line 228). Randomized samples were collected from outside the tents, encompassing both asymptomatic and symptomatic plants. Overall, 14 of 69 plants displayed no leaf symptoms. In order to make things clearer and easily understandable to the reader, we have reworked this part. In this context, we incorporated details regarding the morphological characteristics of the plants (edited document lines 156-159).

  • Comment: In line 228, it is understood that all samples are symptomatic, but it should be stated in the method section that the sampling was made from symptomatic plants.
  • Answer: As previously indicated in comment (a), the requisite information has been incorporated into the Methods section (edited document lines 157–159) to enhance clarity and facilitate reader comprehension in the Results section.

  • Comment: Were the PCR conditions identical for both the fTuf1/rTuf1 and fTufAy/rTufAy primer pairs? I recommend adding the specific cycling parameters for the tuf primers under the subheading in line 179 ('PCR conditions for tuf primers') to ensure methodological transparency. This addition would facilitate experimental reproducibility, particularly given that these primers target the same gene but may require different amplification conditions.
  • Answer: Thank you very much for your insightful comment regarding the PCR conditions used for the primers. We fully agree with you that any deviations in the conditions should be stated to ensure reproducibility and transparency. We used the PCR conditions of Schneider et al. (1997), which is why these conditions were not explicitly mentioned in this study. To answer the question, the PCR conditions were identical for both primers, except for the annealing temperature. The annealing temperatures were 45°C for fTuf1/rTuf1 and 55°C for fTufAy/rTufAy. We included this information in the text (edited document lines 184-186).

  • Comment: The statement in line 194 regarding additional PCR testing of STAMP-positive samples with SecY/VMP1 primers appears redundant. If the stamp gene had been negative, wouldn't verification have been done through vmp1 and secy genes? this sentence should be removed to streamline the text.
  • Answer: We agree with the reviewer's observation that the statement in line 194 (edited document line 196) is redundant. The sentence has been removed.

  • Comment: The heading in line 220 can be corrected as follows; symptomatic distribution in diseased carrots
  • Answer: In accordance with the reviewer’s comment, we have reformulated and enriched the heading in line 220 (edited document line 221) to ‘Symptomatic distribution in diseased carrots’.

  • Comment: Please describe the sex-determination methodology (morphological/molecular) for captured insects in the Methods section. The reported sex ratios (line 245) lack this critical experimental detail.
  • Answer: In the Methods section (edited document lines 126-127) we added information on sex determination: The sex of adult planthoppers was determined based on the anatomy of their external genitalia.

  • Comment: The parenthetical expressions at the end of the figure caption on lines 226 and 252 should be removed or corrected to "photos" instead of "fotos".
  • Answer: In compliance with the reviewer’s comment, we have changed ‘fotos’ to ’photos’ at the end of the figure captions.

  • Comment: While line 245 reports the sex ratio of captured insects, the Methods section lacks details on how males and females were distinguished.
  • Answer: As indicated in the response to comment (h), we added information on sex determination.

  • Comment: Line 267 reports amplification of 'Ca. A. phytopathogenicus' (CaAp) tuf gene via nested PCR, but the Methods section (subsections 2.4.1 and 2.4.3) fails to specify: The primer pairs used for CaAp tuf amplification, Whether these were identical to those used for 'Ca. P. solani' (CaPsol), and If different, the rationale for primer selection. If the same primer pairs successfully amplified both 'Ca. P. solani' (Stol) and 'Ca. A. phytopathogenicus' (CaAp) tuf genes in nested PCR, this finding is particularly noteworthy, as these primers were previously considered Stol-specific.
  • Answer: We agree with the reviewer that it is essential to incorporate significant information regarding analytics in the Methods section. Nonetheless, detection of a tuf amplicon from CaAp amplicon using nested PCR for tuf constitutes a result (edited document line 267). This finding indicates that, similar to CaPsol, CaAp also harbors a tuf gene, although it does not exhibit the same restriction fragment pattern as CaPsol. The tuf primers were well suited for the identification and characterization of CaPsol. However, other bacteria also contain tuf genes, which are likely to be amplified with the primer pairs fTuf1/rTuf1 and fTufAy/rTufAy, yet do either not reveal the CaPsol specific size of the amplification product in the endpoint PCR (940 bp) or show divergent restriction fragment patterns following digestion of the PCR product. Nevertheless, we could show that the tuf gene could be amplified with the tuf-primers employed.

  • Comment: Since the two sentences below Table 1 are explanatory for Table 1, the space between lines 295 and 296 should be removed.
  • Answer: We agree with your observations. The two lines mentioned are explanatory for the selection of the subsamples presented in table 2. We therefore included the information in the Caption of table 2:  “Prevalence of CaAp, CaPsol (16SrXΙΙ-A) and double infection in carrot subsample (N=62) of the 166 samples shown in table 1 with CT values of 10 to 32.5 …” (edited document lines 295-296).

  • Comment: The accession number should be written in parentheses on line 347.
  • Answer: In accordance with the reviewer's suggestion, the Accession number will be added as soon as Acc. No. were generated by the NCBI database on line 344.

  • Comment: A more symmetrical image can be obtained if the dimensions of the two different agarose gel images in line 348 are equalized.
  • Answer: In accordance with the reviewer’s comment, we revised Figure 3 and incorporated the updated version into the edited document (edited document line 345).

  • Comment: On line 357, the genbank accession number should be written in parentheses.
  • Answer: In accordance with the reviewer's suggestion, the Accession number will be added as soon as Acc. No were generated by the NCBI database on line 354.

  • Comment: In Figure 6 in Appendix a, the first well is left blank. A marker should be written on the next well, and the characteristics of this marker should be specified in the figure caption. Furthermore, the in silico gel figure is unnecessarily long; the figure length should be shortened by cutting off the top.
  • Answer: Thank you for your valuable suggestion. We have revised the Figure 6 in Appendix A. It is now shorter and a ladder has been added.

In addition to the above, we have carefully read our paper and have made other changes that improve the quality of the manuscript. We have added a brief conclusion summarizing the findings of this study. This can be seen in lines 497–510.

We want to thank the reviewers again for taking the time to review our paper. We appreciate their helpful comments and feedback, especially their detailed suggestions that improved our paper.

Sincerely yours

Natasha Witczak

Reviewer 2 Report

Comments and Suggestions for Authors

The manuscript entitled “Carrot (Daucus carota L.) as Host for Pentastiridius leporinus and Phloem Restricted Pathogens in Germany” by Natasha Witczak and co-authors investigates whether P. leporinus can infection on carrot. While authors showed that this vector can play critical role in disease development in carrots. Addressing following a few issues would provide clarity to audience:

  1. Authors mention 4 pop-up tents but don't specify whether this number provides its replication types (e.g, biological, technical). Justify sample size
  2. Authors state that plants outside the tents were sampled. Clarify if these are untreated controls??
  3. Authors noted that carrots may now serve as a host. Can you please clarify it in the discussion/conclusion sections to report insect life-cycles
  4. Please provide a detailed description of TaqMan real-time quantitative PCR for pathogen identification including primers and probe (if used)
  5. can you please provide details on the timing of insect development within the 2 months period
  6. I am not sure if there is possibility of host expansion, can you provide explanation if there is such possibility

Author Response

For research article

Response to Reviewer#2 Comments

Sunday, 10.08.2025

Journal: Biology (ISSN 2079-7737)

Manuscript ID: biology-3820481

Type: Article

Title: Carrot (Daucus carota L.) as Host for Pentastiridius leporinus and Phloem Restricted Pathogens in Germany

Authors: Natasha Witczak *, Salma Benaouda, Friederike Wahl, Hendrik Göbbels, Christian Lang, Barbara Jarausch, Michael Maixner

Section: Microbiology

Special Issue: Phytoplasmas: Molecular Characterization and Host–Pathogen Interactions (2nd Edition)

We thank the reviewers for taking the time to read our paper and for their helpful suggestions and comments. These have greatly improved our work. We have carefully addressed all their feedback during the revision. We have explained the changes we made for each comment. Our detailed responses to each point are below.

Answer to Reviewer #2

  • Comment: Authors mention 4 pop-up tents but don't specify whether this number provides its replication types (e.g, biological, technical). Justify sample size
  • Answer: To enhance the clarity and comprehension for the reader, we have specified the type of study (edited document line 110). The number of tents can be considered technical replicates. The study was limited to four tents because of the inability to estimate the number of available planthoppers at time of the experimental setup and ensure an adequate quantity of plant material within the tents.

  • Comment: Authors state that plants outside the tents were sampled. Clarify if these are untreated controls??
  • Answer: We appreciate your insightful comment. Outside the tents, we collected samples to compare and determine whether additional phytoplasmas from the 16SrXII group and CaAp were transmitted to the carrot. This did not constitute an untreated control, as the possibility of additional and/or potential vectors entering the open field cannot be excluded (edited document lines 436-438) and P. leporinus was present also outside the tents. By characterizing the strains that may occur in specialized epidemiological cycles, these serve as a 'fingerprint'. Based on the samples from outside the tents, our objective was to ascertain whether specific combinations in the MLST could allow us to draw conclusions about the vector. We were able to identify the stamp genotype FR662/23 (PP731988) in the samples examined outside the tent, which, until now, has only been detected in sugar beet by Duduk et al. (2023). A specific vector has not been previously described.

  • Comment: Authors noted that carrots may now serve as a host. Can you please clarify it in the discussion/conclusion sections to report insect life-cycles
  • Answer: We have added this to the Discussion section regarding the larval stage (edited document line 416). With this modification, we believe that the information provided clarifies that large parts of the developmental stage of P. leporinus can occur on the carrot. A host plant is characterized by the ability of a pathogen to colonize a plant and of an insect to perform its larval development on this plant. We would like to emphasize that this study is a field trial and that crops cannot remain in the fields indefinitely until the completion of larval development. However, egg deposition and the observation of larval instars up to the fifth (last) instar on the carrot roots suggest that carrot is indeed a suitable host plant for P. leporinus, as already described for sugar beet and potato. In future studies, it would be advisable to investigate the complete life cycle of P. leporinus on carrots under controlled conditions.

  • Comment: Please provide a detailed description of TaqMan real-time quantitative PCR for pathogen identification including primers and probe (if used)
  • Answer: We appreciate your constructive criticism, which has helped us clarify this further. With this in mind, we restructured the sentence (edited document lines 159-161). For pathogen detection, we refer to the protocols described in the respective publications (Therhaag et al., 2024; Zübert & Kube, 2021).

  • Comment: can you please provide details on the timing of insect development within the 2 months period
  • Answer: We are grateful for your constructive comments. Unfortunately, we are not aware of any studies on the development of P. leporinus under field conditions that could answer this question. It is generally known that P. leporinus nymphs go through five larval stages before developing into adults in the following year. To date, the life cycle of P. leporinus has been studied only under laboratory conditions. Under laboratory conditions, development from egg to adult takes 140–170 d (Behrmann et al., 2022; Pfitzer et al., 2022). In our study on carrots, we found nymphs at various stages, indicating a high variation in the duration and stages of development of P. leporinus (edited document lines 248-250). Faster development of some P. leporinus individuals has also been observed in potato (Behrmann et al. 2023; Therhaag et al. 2024). Further research is required to better understand this pest.

  • Comment: I am not sure if there is possibility of host expansion, can you provide explanation if there is such possibility
  • Answer: We appreciate your comment and are pleased to provide a response. The expansion of host plants in the Cixiidae family has been previously documented. Maixner et al. (2014) demonstrated the host expansion of Hyalesthes obsoletus to Urtica dioica L. In the case of P. leporinus, the transition from reed to sugar beet was initially observed and described in France (Bressan et al., 2010). In 2023, Behrmann et al. (2023) identified P. leporinus in potatoes. These observations have been corroborated in various countries (EU). It appears that these species exhibit a preference for crop plants with underground storage organs, which provide optimal nourishment for nymphs and facilitate their development.

In addition to the above, we have carefully read our paper and have made other changes that improve the quality of the manuscript. We have added a brief conclusion summarizing the findings of this study. This can be seen in lines 497–510.

We want to thank the reviewers again for taking the time to review our paper. We appreciate their helpful comments and feedback, especially their detailed suggestions that improved our paper.

Sincerely yours

Natasha Witczak

Reviewer 3 Report

Comments and Suggestions for Authors

The manuscript is well written, clearly structured, and presents an important and timely study on the role of Pentastiridius leporinus in transmitting phloem-restricted pathogens to carrots. The experimental design is strong, and the findings offer new insights that are valuable for understanding disease spread in carrot cultivation. The corrections made in the revised version have improved the clarity, accuracy, and overall flow of the text. Scientific terms and names have to use correctly, and the changes need careful attention to detail. Overall, this is a well-executed and clearly presented study.

Author Response

For research article

Response to Reviewer#3 Comments

Sunday, 10.08.2025

Journal: Biology (ISSN 2079-7737)

Manuscript ID: biology-3820481

Type: Article

Title: Carrot (Daucus carota L.) as Host for Pentastiridius leporinus and Phloem Restricted Pathogens in Germany

Authors: Natasha Witczak *, Salma Benaouda, Friederike Wahl, Hendrik Göbbels, Christian Lang, Barbara Jarausch, Michael Maixner

Section: Microbiology

Special Issue: Phytoplasmas: Molecular Characterization and Host–Pathogen Interactions (2nd Edition)

We thank the reviewers for taking the time to read our paper and for their helpful suggestions and comments. These have greatly improved our work. We have carefully addressed all their feedback during the revision. We have explained the changes we made for each comment. Our detailed responses to each point are below.

Answer to Reviewer #3

  • Comment: Line 33 (Daucus carota): It should be Daucus carota
  • Answer: As suggested by the reviewer, we revised the scientific name.

  • Comment: Line 56 (Allium cepa): It should be Allium cepa
  • Answer: We agree with the reviewer’s comment. Indeed, the scientific name has been changed.

  • Comment: Line 68 (de novo): The text should be italicized
  • Answer: We appreciate your insightful comments on line 68 (de novo). We agree with your suggestion that this term should be italicized as it is a foreign word. Accordingly, in the revised version of the document, we adjusted the font in line 68 to reflect this change.

  • Comment: Line 72 (high crop losses in carrot): What is the percentage?
  • Answer: We are grateful for your constructive comment. The cited source does not provide exact figures, which is why we refrained from specifying percentages. In the literature, harvest losses of up to 80% are reported (Plant Pathology (5th Edition), 2005), but these figures refer to ‘Ca. Phytoplasma asteris’ (16SrI) and cannot be directly applied to 16SrXII phytoplasmas. With this study, we aim to raise awareness of this issue and hope that such important data related to 16SrXII phytoplasma infection will be collected and published in the coming years.

  • Comment: Line 80: In the field
  • Answer: Thank you for this improvement. This helped us to enhance the English in the manuscript. The correction has been made (edited document line 80).

  • Comment: Line 109 (2.2 Field Collection of leporinus and Transmission Experiment): Change capital letter to small letter
  • Answer: In compliance with the reviewer’s comment, we have changed the capital letters to small letters in Section 2.2.

  • Comment: Line 226/252 ((Fotos)): Replace all instances of ‘Fotos’ with ‘Photos’
  • Answer: Similarly as in “Answer to Reviewer #1 (g)” we changed ‘fotos’ into ‘photos’ (edited document line 227 and 253).

  • Comment: Line 247 removed text ‘inoculation access period’
  • Answer: To make things clearer and more understandable to the reader, we have decided to spell out the abbreviation IAP at the beginning (edited document line 119), as it is a specific abbreviation that cannot be assumed to be ‘common knowledge’.

  • Comment: Line 357 (Acc. No. XXXX): ???
  • Answer: We apologize for this confusion. The exes were intended to be placeholders to avoid forgetting the Accession numbers. The Accession number will be added as soon as Acc. No were generated by the NCBI database on line 354.

  • Comment: Line 404 (Beta vulgaris): Beta vulgaris ssp. vulgaris
  • Answer: As suggested by the reviewer, we revised the scientific name.

  • Comment: Line 404 (Solanum tuberosum): Solanum tuberosum
  • Answer: As suggested by the reviewer, we revised the scientific name.

  • Comment: Line 524 (Figure 6): ??
  • Answer: In accordance with the reviewer’s comment, we have rephrased and expanded the title of Figure 6. Specifically, we have added ‘16SrXII’ to clarify that it refers to other strains of the 16SrXII group (edited document line 543).

In addition to the above, we have carefully read our paper and have made other changes that improve the quality of the manuscript. We have added a brief conclusion summarizing the findings of this study. This can be seen in lines 497–510.

We want to thank the reviewers again for taking the time to review our paper. We appreciate their helpful comments and feedback, especially their detailed suggestions that improved our paper.

Sincerely yours

Natasha Witczak
